# Social media use and adolescent sleep patterns: cross-sectional findings from the UK millennium cohort study

Holly Scott, Stephany M Biello, Heather Cleland Woods

This work presents secondary analysis of data deposited in the UK Data Service. It is therefore not possible for the authors to share the current findings directly with original study participants.

School of Psychology, University of Glasgow, Glasgow, UK

**Correspondence to**
Holly Scott;
h.scott.1@research.gla.ac.uk

## ABSTRACT

**Objectives** This study examines associations between social media use and multiple sleep parameters in a large representative adolescent sample, controlling for a wide range of covariates.

**Design** The authors used cross-sectional data from the Millennium Cohort Study, a large nationally representative UK birth cohort study.

**Participants** Data from 11 872 adolescents (aged 13–15 years) were used in analyses.

**Methods** Six self-reported sleep parameters captured sleep timing and quality: sleep onset and wake times (on school days and free days), sleep onset latency (time taken to fall asleep) and trouble falling back asleep after nighttime awakening. Binomial logistic regressions investigated associations between daily social media use and each sleep parameter, controlling for a range of relevant covariates.

**Results** Average social media use was 1 to <3 hours per day (31.6%, n=3720). 33.7% were classed as low users (<1 hour; n=3986); 13.9% were high users (3 to <5 hours; n=1602) and 20.8% were very high users (5+ hours; n=2203). Girls reported spending more time on social media than boys. Overall, heavier social media use was associated with poorer sleep patterns, controlling for covariates. For example, very high social media users were more likely than comparable average users to report late sleep onset (OR 2.14, 95% CI 1.83 to 2.50) and wake times (OR 1.97, 95% CI 1.32 to 2.93) on school days and trouble falling back asleep after nighttime awakening (OR 1.36, 95% CI 1.10 to 1.66).

**Conclusions** This study provides a normative profile of UK adolescent social media use and sleep. Results indicate statistically and practically significant associations between social media use and sleep patterns, particularly late sleep onset. Sleep education and interventions can focus on supporting young people to balance online interactions with an appropriate sleep schedule that allows sufficient sleep on school nights.

## INTRODUCTION

There is significant current attention towards the possible impact of screen time and social media on our adolescents' health. However, the lack of empirical evidence to support policy and practice development in this area has been consistently voiced by clinicians and researchers. For example, at the UK House of Commons Science and Technology Committee inquiry into the impact of social media and screen use on young people's health use in adolescence, the Royal College of Paediatrics and Child Health (RCPCH) urged the UK government as a matter of priority to develop guidance for health practitioners along the same lines as the American Academy of Paediatrics (AAP) but importantly based on UK data.[1 2] They also argue along with other researchers that there is a need to refocus away from correlations between generic terms such as 'screentime' and poor well-being, towards meaningfully quantifying how various types of technology use impact on different areas of child and adolescent health and well-being. This study presents UK data that provide a nationally representative profile of current adolescent social media use and takes a data-driven approach to quantify sleep patterns for high and very high users relative to average users.

This study focuses on sleep, which—despite often being overlooked in public health messages and education interventions[3 4]—is increasingly recognised as a key component of wider health and well-being.[5] Adolescent sleep is an important public health issue, as insufficient sleep is highly prevalent in this age group and has implications for mental health, obesity, academic performance and safety.[6] With the majority of adolescents reporting

insufficient sleep to function properly or to meet recommended guidelines,[7 8] there is growing concern that social media may be a contributing factor for today's teenagers. For example, the potential for 24/7 social media interactions may exacerbate the existing conflict of early school start times with naturally delayed adolescent rhythms and other social and educational demands.[6 9 10] As a highly relevant issue for paediatric practice, there is a clear need for UK evidence to inform and update decision-making in medical practice and policy to address this current issue in adolescent sleep.

This study responds to this need, presenting large-scale UK data on adolescent sleep and social media use, while addressing a number of existing gaps in available international evidence. In addition to providing much-needed UK evidence, the current approach also addresses the need for evidence that (1) examines social media specifically, rather than generic screentime; (2) isolates effects for a range of sleep parameters by accounting for an extensive range of covariates and (3) frames these effects within the context of current adolescent social media norms to provide meaningful comparisons. The current approach to ensure that this evidence can meaningfully inform policy and practice by addressing each of these needs is discussed further below.

First, it is important for available evidence to examine social media individually, rather than aggregating these interactions and other media use under the umbrella term 'screentime'. A recent large-scale US study indicated a significant but modest effect for overall screentime and sleep and called for future research to examine effects for specific technologies.[11] In particular, the interactive nature of social media presents uniquely relevant issues for adolescent sleep compared with other forms of screentime or traditional media.[7 12] Although facilitated by screens, social media interactions are underpinned by the same drivers as any social interaction, with a desire for inclusion and belonging mixing with concerns over violating social expectations or etiquette.[13] These concerns can make it difficult to disengage from social media at bedtime, with some adolescents identifying this as a cause of delayed sleep onset and daytime tiredness.[13] These unique social and emotional aspects of online interactions underline the importance of examining social media use specifically, in relation to adolescent sleep outcomes.

Second, to meaningfully inform an evidence-based response to social media use, research must examine multiple sleep parameters and isolate these effects by controlling for an extensive range of relevant covariates. This is crucial to assess the practical significance of underlying direct effects[11] and to identify which aspects of adolescent sleep merit attention to social media use in practice and policy. Available research on social media use and sleep is often left questioning whether reported effects could be explained by other individual factors: for example, if more anxious, depressed or sedentary adolescents may tend to both use social media more

and report poorer sleep.[12 14] Individual studies that have controlled for specific groups of covariates generally suggest that associations do persist.[15 16] However, there remains a need for large-scale evidence that addresses a wide range of covariates simultaneously, to more robustly establish which dimensions of sleep have a direct association with social media use and which reflect another underlying issue (eg, anxious or depressive symptoms).

This type of evidence is required to invest time and resources effectively, by identifying which sleep complaints may benefit from directly addressing social media use. For example, a range of sleep complaints from insufficient sleep to problems initiating or maintaining sleep have been examined in relation to social media use. In terms of sleep duration, time spent using social media may displace sleep directly or displace other daytime activities (such as homework) that are then delayed and disrupt nighttime routines.[9 17] Social media use may also impact on the quality of sleep via increased arousal, not simply through light exposure,[18] but particularly via cognitive and social activity.[12 14 19] Given these different potential mechanisms, it is therefore important to examine social media use in relation to a range of sleep parameters, to identify which of these links have the most practical significance after accounting for relevant factors.

Third, evidence should frame these effects within the context of current norms for adolescent social media use. Research to date has tended to focus on problematic or 'addicted' social media users[10 20] or to compare outcomes for the highest users against the lowest users.[15 17 21] In contrast, first establishing what constitutes typical use and then comparing outcomes for relatively higher or lower users against this reference point can support more meaningful conclusions. This data-driven approach avoids imposing arbitrary or quickly outdated cut-offs, taking into account recent rapid increases in social media use.[22] Comparing sleep patterns for higher users against average users can better support practical and realistic discussions on best practice that consider the context of current adolescent social media norms.

This study targets these existing gaps in available international evidence, while providing much-needed large-scale UK evidence. It examines associations between social media use and multiple sleep parameters in a large, nationally representative adolescent sample: the UK Millennium Cohort Study.[23] It first investigates current norms in adolescent social media use to establish the average level of daily use and the prevalence of comparatively high use. It then examines which sleep parameters are associated with social media use, isolating these effects by controlling for extensive covariates and quantifying effects for higher users relative to average users. This aims to provide rigorous and meaningful evidence to inform practice and policy to support healthy adolescent sleep and social media use.

## METHODS

### Participants

The UK Millennium Cohort Study is a nationally representative, multidisciplinary survey which aims to explore the influence of family context on child and adolescent development and outcomes. The survey covers a broad range of domains, such as parenting, housing, poverty and health. It consists of a random two-stage sample drawn from all live UK births in the 12-month period starting 1 September 2000 in England and Wales and 1 December 2000 in Scotland and Northern Ireland, identified through the Child Benefit register.[23] The clustered sample is drawn from a disproportionately stratified sample of electoral wards (local areas) to provide adequate representation of areas with higher concentrations of minority ethnic and disadvantaged families. Parents completed the first survey sweep in 2001 when their child was aged 9 months, with 18 818 cohort members. Children also completed surveys from age 7 (sweep 4) onwards. The most recent survey (sweep 6 at around age 14) gathered self-report data from 11 872 cohort members, including questions on their typical social media use and sleep patterns. Parents were required to give written consent to complete the parent survey and for the interviewer to invite the cohort member to participate in the young person survey. Cohort members then also had to give verbal consent to complete the young person survey, which was self-completed on the interviewer's tablet.

### Materials

The current analyses make use of available data from the UK Millennium Cohort Study, which measured social media use and sleep using single-item self-report questions. Although not validated questionnaire measures, these survey questions do provide a snapshot of the subjective experience of sleep and social media use in this large representative sample, capturing a range of sleep habits and the typical time spent using social media each day.

### Social media use

Participants indicated how much time they spent using social media on a typical weekday, choosing from eight response categories (ranging from 0 hours to 7+ hours) to answer the following question: 'On a normal week day during term time, how many hours do you spend on social networking or messaging sites or Apps on the internet such as Facebook, Twitter and WhatsApp?'

### Sleep parameters

Participants reported typical sleep habits through six single items (each with five or six response categories) that assessed: sleep onset and wake times (on school days and free days, separately), sleep onset latency (time taken to fall asleep) and trouble falling back asleep after night-time awakening. The online supplementary materials provide a full list of items and response categories.

### Covariates

In addition, the following relevant covariates (identified based on literature) had available data in the UK Millennium Cohort Study: demographics (ethnic minority status, Organisation for Economic Cooperation and Development equivalised weekly family income); family composition (number of siblings in household, the presence of both parents, age of primary parent/carer responder); psychosocial adjustment (using the parent-report Strengths and Difficulties Questionnaire)[24]; depressive symptoms (using the Short Mood and Feelings Questionnaire)[25]; self-esteem (using a shortened and adapted version of the Rosenberg Self-Esteem Scale)[26] and general health (single item), social support (three items) and physical activity (single item).

### Data analysis

Since the aim of the study was to compare sleep outcomes for high and low users vs average users, based on the distribution we initially collapsed responses into three categories: under 1 hour for 'low' users (33.7%), 1 to <3 hours for 'average' users (31.6%) and 3 hours or more for 'high' users (34.7%). Given the broad range covered by this 'high' user category (including responses of 3 to <5 hours, 5 to <7 hours and 7+ hours), and with sufficient numbers, we separated this into 'high' (3 to <5 hours; 13.9%) and 'very high' users (5+ hours; 20.8%) to allow more detailed exploration.

We collapsed responses for each sleep measure into binary outcomes. For poor sleep quality, these outcomes were: sleep onset latency over 30 min (commonly used to indicate poor sleep quality)[27 28] and difficulty falling asleep following nighttime awakenings at least 'a good bit of the time'. For late sleep onset and wake times, we took a data-driven approach to identify meaningful cut-off points, which were defined as later than average (including responses in categories later than the median response category). The table 1 summarises the resulting criteria for each sleep outcome and associated prevalence rates.

Separate binomial logistic regression models predicted ORs of each sleep outcome for low, high and very high social media users, compared with average users. We ran models that controlled only for exact age and sex, followed by models that further controlled for measures of demographics, family characteristics, psychological well-being and health (see the Materials section for full details). All analyses allowed for the complex survey design (with its clustered, stratified sample) and used longitudinal weights to account for non-random longitudinal attrition from the sample, using the 'survey' and 'srvyr' packages in R.[29–31]

Multiple imputation was performed to account for missing data, reducing bias and increasing power.[32] The overall missing data rate was 2.8%, ranging from 0.0% to 6.0% for individual measures, with most measures below 5%. We make the assumption that data are missing at random (ie, that patterns of missingness can be explained

**Table 1** Social media use and sleep outcomes: criteria and prevalence

| Variable | Criteria | Prevalence (%) | | |
|---|---|---|---|---|
| | | Male | Female | Total |
| Daily social media use | | | | |
| Low | <1 hour | 43.8 | 22.8 | 33.7 |
| Average | 1 to <3 hours | 32.1 | 31.1 | 31.6 |
| High | 3 to <5 hours | 10.4 | 17.7 | 13.9 |
| Very high | 5+ hours | 13.7 | 28.4 | 20.8 |
| Sleep outcomes | | | | |
| Late sleep onset (school day) | After 23:00 | 25.5 | 26.5 | 26.0 |
| Late sleep onset (free day) | After midnight | 35.2 | 32.1 | 33.7 |
| Late wake time (school day) | After 08:00 | 5.3 | 2.7 | 4.0 |
| Late wake time (free day) | After 11:00 | 22.5 | 21.5 | 22.0 |
| Long sleep onset latency | Over 30 min | 31.5 | 37.0 | 34.1 |
| Trouble falling back asleep after nighttime awakening | At least 'a good bit of the time' | 16.9 | 25.7 | 21.1 |

Percentages account for survey design and weights. Criteria for late sleep onset and wake times defined as later than the median response category. Gender difference in daily social media use, late wake time (school day), long sleep onset latency and trouble falling back asleep after nighttime awakening p<0.001; gender difference in late sleep onset (free day) p<0.01. For a breakdown of social media use by other demographics (household income and ethnicity), see the online supplementary materials.

by other variables available in the data).[32] All variables, including covariates, were used in the imputation model, which was run using R package 'mice'.[33] Estimates were combined across 10 imputed data sets (each produced through 10 iterations). Results were similar for analyses on multiply imputed and complete case data, so only multiply imputed analysis is presented here.

## RESULTS

The median time spent using social media on a typical day was 1 to <3 hours (32% of adolescents); however, 21% used social media for at least 5 hours. Girls tended to use social media more than boys (see table 1).

Median sleep onset times were 22:00–23:00 on school days (with 26% falling asleep later than this) and between 23:00 and midnight on free days (with 34% falling asleep later; see table 1). Median wake times were 07:00–08:00 on school days (with only 4% waking later than this) and 10:00–11:00 on free days (with 22% waking later). Boys were more likely to fall asleep late on free days and wake up late on school days. In measures of poor sleep quality, 34% typically took longer than 30 min to fall asleep and 21% reported difficulties falling asleep following nighttime awakenings at least 'a good bit of the time'. Girls were more likely to have long sleep onset latency and trouble falling back asleep after nighttime awakening.

Separate binomial logistic regression models explored whether odds of each sleep outcome differed for low, high and very high social media users, compared with average users (1 to <3 hours). First, models controlled for exact age and sex (see table 2). Very high social media use (5+ hours) was associated with higher odds of all six sleep outcomes. High social media use (3 to <5 hours) was associated with higher odds of all outcomes except for late rise times on free days. Low social media use (<1 hour) was associated with lower odds of late sleep onset on school days and free days and late wake times on free days.

Further modelling then controlled for a more comprehensive set of covariates (see table 3, with note detailing list of covariates). High social media use was no longer significantly associated with long sleep onset latency or frequent nighttime awakenings. Very high social media was no longer significantly associated with long sleep onset latency; however, its association with frequent nighttime awakenings remained significant but smaller. Patterns of significant associations for late sleep onset and wake times remained unchanged, although effect sizes were reduced, particularly for very high social media use.

For ease of interpretation, we also transformed the resulting adjusted ORs from these covariate models into adjusted relative risks[34] (see table 4). These summarise differences in *probabilities,* as opposed to odds, and can be interpreted more intuitively. For example, the adjusted relative risk of 1.68 indicates that an adolescent with very high social media use is 68% more likely to fall asleep after 11pm on school nights than a comparable adolescent (controlling for covariates) with average social media use.

## DISCUSSION

This study aimed to address calls from those working in policy and practice to establish a UK data-driven profile of current adolescent daily social media use and to examine links to a key component of wider adolescent health and well-being using multiple sleep parameters

**Table 2** Binomial logistic regressions (adjusting only for age and sex)

| | Low: <1 hour | | High: 3 to <5 hours | | Very high: 5+ hours | |
| --- | --- | --- | --- | --- | --- | --- |
| | OR (95% CI) | P value | OR (95% CI) | P value | OR (95% CI) | P value |
| Late sleep onset (school days) | 0.63 (0.53 to 0.75) | <0.001 | 1.38 (1.16 to 1.65) | <0.001 | 2.75 (2.38 to 3.18) | <0.001 |
| Late sleep onset (free days) | 0.6 (0.51 to 0.7) | <0.001 | 1.44 (1.19 to 1.74) | <0.001 | 3.05 (2.63 to 3.54) | <0.001 |
| Late wake time (school days) | 1.12 (0.77 to 1.62) | ns | 1.63 (1.05 to 2.53) | <0.05 | 2.49 (1.62 to 3.83) | <0.001 |
| Late wake time (free days) | 0.81 (0.69 to 0.96) | <0.05 | 1.16 (0.97 to 1.39) | ns | 1.82 (1.54 to 2.16) | <0.001 |
| Sleep onset latency >30 min | 0.92 (0.8 to 1.05) | ns | 1.24 (1.04 to 1.49) | <0.05 | 1.48 (1.27 to 1.71) | <0.001 |
| Frequent nighttime awakenings | 1.06 (0.9 to 1.26) | ns | 1.31 (1.07 to 1.61) | <0.05 | 2.11 (1.75 to 2.55) | <0.001 |

See table 1 for criteria and prevalence of each sleep outcome. ORs measure how much higher or lower the odds of a given sleep outcome are for each category of social media user (low: <1 hour; high: 3 to <5 hours; very high: 5+ hours) compared with the reference category (average users: 1 to <3 hours). ORs and 95% CIs greater than 1 indicate higher odds; those below 1 indicate lower odds. ORs are adjusted to control for exact age and sex.
ns, not significant.

while accounting for a wide range of covariates, using data from a large nationally representative sample of UK adolescents. The results highlighted a wide range of reported daily social media use, with tertiles defining low, average and high use on a typical school day as <1 hour, 1 to <3 hours and 3+ hours, respectively. This indicates generally heavier social media use compared with young adults[21] and provides a current normative profile for UK adolescents. One in five adolescents were classed as very high users, spending 5+ hours using social media on a typical school day, whereas two-thirds of the sample used social media for less than 3 hours. This provides a data-driven profile of use to support decision-making, rather than relying on assumptions around prevalence of high use. In line with previous studies, girls tended to spend more time on social media than boys[35 36] and report poorer sleep quality.[37 38] This reinforces the importance of controlling for gender when examining these associations and highlights the need for continued work to explore the sleep implications of *how* adolescent boys and girls spend their time on social media (with previous evidence of gender differences in preferred platforms, motivations and self-presentation).[35 36 39]

In terms of sleep timing, social media use remained significantly associated with late sleep onset and wake times after controlling for covariates, with the strongest

**Table 3** Binomial logistic regressions (with further adjustments for covariates)

| | Low: <1 hours | | High: 3 to <5 hours | | Very high: 5+ hours | |
| --- | --- | --- | --- | --- | --- | --- |
| | OR (95% CI) | P value | OR (95% CI) | P value | OR (95% CI) | P value |
| Late sleep onset (school days) | 0.61 (0.51 to 0.73) | <0.001 | 1.23 (1.02 to 1.49) | <0.05 | 2.14 (1.83 to 2.5) | <0.001 |
| Late sleep onset (free days) | 0.57 (0.49 to 0.68) | <0.001 | 1.32 (1.09 to 1.6) | <0.01 | 2.41 (2.08 to 2.79) | <0.001 |
| Late wake time (school days) | 1.04 (0.71 to 1.5) | ns | 1.56 (1.02 to 2.4) | <0.05 | 1.97 (1.32 to 2.93) | <0.01 |
| Late wake time (free days) | 0.79 (0.67 to 0.93) | <0.01 | 1.12 (0.92 to 1.35) | ns | 1.57 (1.32 to 1.87) | <0.001 |
| Sleep onset latency >30 min | 0.9 (0.78 to 1.04) | ns | 1.11 (0.92 to 1.34) | ns | 1.12 (0.96 to 1.32) | ns |
| Frequent nighttime awakenings | 1.04 (0.88 to 1.24) | ns | 1.08 (0.87 to 1.35) | ns | 1.36 (1.1 to 1.66) | <0.01 |

See table 1 for criteria and prevalence of each sleep outcome. Reference category is average (1 to <3 hours). ORs are adjusted to control for: exact age, sex, ethnic minority status, family income, number of siblings in household, the presence of both parents in household, parent's age, Strengths and Difficulties score, Mood and Feelings score, self-esteem, general health, social support and physical activity.
ns, not significant.

**Table 4** RR (from covariate-adjusted models)

| | Low: <1 hour | | High: 3 to <5 hours | | Very high: 5+ hours | |
|---|---|---|---|---|---|---|
| | RR (95% CI) | P value | RR (95% CI) | P value | RR (95% CI) | P value |
| Late sleep onset (school days) | 0.67 (0.58 to 0.78) | <0.001 | 1.17 (1.02 to 1.33) | <0.05 | 1.68 (1.52 to 1.84) | <0.001 |
| Late sleep onset (free days) | 0.66 (0.58 to 0.75) | <0.001 | 1.2 (1.06 to 1.35) | <0.01 | 1.69 (1.57 to 1.81) | <0.001 |
| Late wake time (school days) | 1.03 (0.72 to 1.48) | ns | 1.54 (1.02 to 2.29) | <0.05 | 1.91 (1.3 to 2.76) | <0.01 |
| Late wake time (free days) | 0.83 (0.72 to 0.95) | <0.01 | 1.09 (0.94 to 1.26) | ns | 1.41 (1.24 to 1.59) | <0.001 |
| Sleep onset latency >30 min | 0.93 (0.84 to 1.03) | ns | 1.07 (0.94 to 1.21) | ns | 1.08 (0.97 to 1.2) | ns |
| Frequent nighttime awakenings | 1.03 (0.89 to 1.19) | ns | 1.07 (0.89 to 1.28) | ns | 1.28 (1.09 to 1.5) | <0.01 |

RR transformed from ORs in covariate-controlled binomial logistic regression models (see table 3). Reference category is average (1 to <3 hours). For example, RR of 1.68 means very high users are 68% more likely to fall asleep late on school days than comparable average users.
RR, relative risks; ns, not significant.

effect for sleep onset. Very high social media users were roughly 70% more likely than comparable average users to fall asleep later than average, that is, after 23:00 on school days and after midnight on free days. Low social media users were least likely to fall asleep late, indicating that unlike mental well-being, optimal outcomes for sleep are associated with minimal—not moderate—use.[40] These findings are consistent with the idea that social media displaces sleep: either directly or indirectly.[9][17] Direct sleep displacement may be particularly likely on school days, especially for very high users, since limited social media access during school hours means that at least part of this daily time on social media is likely to take place close to bedtime. Bedtime social media use can delay sleep onset,[14] with some adolescents reporting difficulties disengaging from social media to sleep.[13] A similar process could also indirectly delay sleep onset, if other daytime activities (eg, homework) are delayed due to a sense of urgency to check and respond to social media notifications. This link to later sleep onset is a particular concern on school days, as late school day bedtimes longitudinally predict poorer academic and emotional outcomes.[41] While the survey question aimed to measure sleep onset time by asking what time participants 'go to sleep', some participants may have reported the time that they get into bed, in which case actual sleep onset would be even further delayed.[42]

Social media use was also associated with later wake times on school days (for both high and very high users) and on free days (for very high users). This overall pattern of later sleep timing among heavier social media users could be driven partly by circadian factors, if adolescents with a natural preference for later sleep timing use social media to fill time in the late evening until they feel sleepy. This possibility merits further investigation. Alternatively, this later sleep timing could suggest that heavier

social media users may compensate for later sleep onset with later wake times that still allow sufficient sleep. This compensation may be possible on free days, with flexible rise times. However, on school days only 4% of adolescents reported late wake times (after 08:00), as fixed rise times mean that later sleep onset effectively equates to shorter sleep opportunity on school days.[6][41] Consequently, these slightly later rise times are unlikely to fully compensate for delayed sleep onset on school days and suggest sleep restriction in a population where sleep need is high.[6] Across the sample, this observed pattern of later sleep onset and rise times on free days compared with school days is consistent with well-established delays to the circadian rhythm during this developmental period,[43][44] with growing pressure on policymakers to delay school start times to better align with adolescent body clocks.[45]

Delayed sleep onset is therefore a key issue to target in relation to adolescents' social media use. The current cross-sectional study cannot establish causality; however, some adolescents do report delaying bedtimes as a result of social media use.[13][46] Adolescent sleep interventions should therefore consider assessing the impact of social media use on sleep schedules as standard. Further research can explore adolescents' motivations for prioritising social media over other needs, including sleep,[13] and identify factors that lead some individuals to struggle with this more than others. This can inform efforts to effectively support young people to balance online interactions - and the benefits they can offer[40][47][48] - with an appropriate and consistent sleep schedule across the week, particularly to allow sufficient sleep on school nights. By helping to combat insufficient sleep, this can have a positive impact on adolescent physical and mental health, daytime functioning and academic performance, addressing a significant health and educational burden.[6]

In terms of sleep quality, very high social media users were more likely to experience nighttime awakenings than comparable average users, whereas the effect for long sleep onset latency was fully explained by covariates. Previous studies have found a significant association between social media use and measures of sleep disturbance (including long sleep onset latency and difficulty falling asleep) when controlling for: age and sex[19]; sociodemographic measures[21] and sleep hygiene behaviours.[15] The current more extensive set of covariates also included measures of psychological well-being (depression and psychosocial adjustment), which were strong predictors of long sleep onset latency and have been shown to be linked to generic screentime in previous work.[40] Therefore, considering previous and current findings together, this suggests that although adolescents who spend more time on social media do tend to take longer to fall asleep, both these behaviours could reflect underlying aspects of well-being, with depression and anxiety linked to both poor sleep quality and social media use.[16] This is consistent with evidence that sleep onset latency and presleep cognitive arousal is predicted by underlying concerns about potentially missing out, rather than social media behaviour itself.[14] Since the purpose of this study was to isolate and quantify associations between social media use and sleep, the current approach of including well-being measures as covariates provided this insight into which sleep associations do and do not persist independent of well-being and other covariates. However, future studies can specifically examine in more detail which aspects of mental health and well-being may mediate or moderate these associations. Given the increasing recognition of sleep and mental health as two inextricably linked aspects of health,[49] the current findings lay the foundation for more complex model testing to examine the likely bidirectional and interactive effects between social media use, sleep, mental health and other associated measures, such as school performance. Applying this approach to longitudinal and experimental data will be particularly valuable to elucidate these complex mechanisms and to build a more holistic and balanced understanding of social media's links to both positive and negative aspects of health and well-being.

In contrast, the association between social media use and nighttime awakenings was only partly explained by covariates, with very high social media users still 28% more likely to have frequent difficulties with nighttime awakenings than comparable average users. Social media notification alerts may disrupt sleep during the night, particularly if users then respond by re-engaging with social media. Adolescents who use social media more also tend to have a stronger emotional connection to platforms and experience more fear of missing out.[14 50] Therefore, it is possible that very high users are more likely to remain vigilant for incoming social media alerts or to respond to these during the night, increasing arousal and contributing to difficulties falling asleep again. Further research can focus on this type of specific social media behaviours during the night, to examine whether they explain the link between higher overall use and nighttime awakenings. If incoming alerts are indeed mostly responsible, interventions can promote simple practical steps such as setting 'do not disturb' periods on social media apps.

## Limitations

These findings should be considered within the limitations of the current study. Given the broad scope of the UK Millennium Cohort Study, sleep and social media use were measured using individual questions rather than validated multi-item questionnaires. This limits the current analyses to a single measure of social media use—defined as the amount of time spent using social media on a typical day—which does not capture the different experiences of individual users,[51] for example, in terms of content, context, timing and emotional engagement. Future research should carefully consider a range of measures to provide a more holistic view of adolescents' experiences of using social media, particularly since evidence highlights the importance of emotional and cognitive aspects of social media use for sleep.[14 16] To support future research, there is a clear need to establish *validated* measurement tools that move beyond hours per day to capture these more nuanced aspects of social media engagement. This is a key area for future development, as available tools limit the scope of potential research questions, conclusions and recommendations. Improved measurement tools moving forward can enhance understanding of the mechanisms linking social media use and sleep, as well as providing a more balanced view of both positive and negative impacts of social media experiences.

These analyses make use of six available reported sleep parameters from this representative cohort, which do allow a rounded picture of timing and quality, but future research would benefit from including validated measures of sleep duration and quality, as well as circadian preference. The current self-reported sleep measures offer valuable insight into adolescents' *subjective experience* of sleep, which is one important component of sleep, but this can diverge from verifiable objective measures of sleep parameters provided by other methods, such as the gold-standard polysomnography.[52 53] In particular, sleep state misperception in both good and poor sleepers can result in poor self-report estimates of sleep onset latency,[53] although these differences in sleep onset latency estimates from self-report and polysomnography tend to be small.[54] The current analyses therefore contribute one part of the picture, with a continued need to triangulate insight from multiple methodologies (both subjective and objective) to build a more nuanced, holistic understanding of adolescent social media use and sleep.[52 55]

Furthermore, this study presents cross-sectional data, which precludes conclusions of causality. Cross-sectional analyses are prevalent in this research area, with calls for longitudinal and experimental work to enrich current understanding.[56 57] Recent studies on general technology use have supported bidirectional links between higher technology use and shorter sleep in adolescents,[58] with evidence

that restricting phone access can advance bedtimes and extend sleep opportunity.[59] Restricting adolescent social media use is likely to be especially challenging, as this is a developmental period of increasing autonomy during which peer interactions, belonging and acceptance are highly valued.[2] Therefore, an important avenue for future research is establishing how best to support young people to balance these rewarding online social interactions with an appropriate and consistent sleep schedule, to optimise associated health and school outcomes.

Finally, we note that research in this area is constantly contending with rapidly evolving social media platforms and associated norms and expectations for online interactions. This can be particularly challenging for this type of large national cohort data, which in this case provides a snapshot of UK adolescents' social media use in 2015.

## CONCLUSIONS

This study provides robust evidence on associations between social media use and sleep outcomes, controlling for an extensive range of covariates, in a large nationally representative sample of UK adolescents. It provides a normative profile of adolescent social media use and sleep in the UK, which can be used as a baseline to support evidence-based decision-making policy and practice rather than relying on assumptions around prevalence of high social media use. The findings indicate statistically and practically significant associations between social media use and sleep patterns, particularly late sleep onset. Future research should explore the context and experience of time using social media to inform more meaningful discussions around best practice and updating sleep education and interventions to meet the needs of today's society. Interventions should focus on addressing delayed sleep onset, by supporting young people to balance online social interactions with an appropriate and consistent sleep schedule that allows sufficient sleep on school nights, with benefits for health and educational outcomes.

**Acknowledgements** The authors would like to thank A Przybylski and J Lewsey for helpful discussions when planning methodology.

**Contributors** HS designed the study and carried out data analysis. HS, HCW and SMB interpreted the findings. HS drafted the manuscript in consultation with HCW. HCW and SMB revised the manuscript for important intellectual content. All authors approve the submitted manuscript and agree to be accountable for all aspects of the work.

**Funding** This study was funded by an Economic and Social Research Council +3 PhD studentship for HS (grant number ES/J500136/1). The Millennium Cohort Study was funded by the Economic and Social Research Council.

**Competing interests** All authors have completed the ICMJE uniform disclosure form at www.icmje.org/coi_disclosure.pdf and declare no support from any organisation for the submitted work; no financial relationships with any organisations that might have an interest in the submitted work in the previous three years; no other relationships or activities that could appear to have influenced the submitted work.

**Patient consent for publication** Not required.

**Ethics approval** The Millennium Cohort Study Sweep 6 was approved by the London Multicentre Research Ethics Committee (13/LO/1786).

**Provenance and peer review** Not commissioned; externally peer reviewed.

**Data availability statement** Data are available in a public, open access repository.

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
