## [Reviewer comments · BMJ Open]

This paper was submitted to a another journal from BMJ but declined for publication following peer review. The authors addressed the reviewers' comments and submitted the revised paper to BMJ Open. The paper was subsequently accepted for publication at BMJ Open.

ARTICLE DETAILS

TITLE (PROVISIONAL)	Social media use and adolescent sleep patterns: cross-sectional findings from the UK Millennium Cohort Study
AUTHORS	Scott, Holly; Biello, Stephany M; Woods, HC

VERSION 1 – REVIEW

REVIEWER	Shahrad Taheri Weill Cornell Medicine I am on advisory board for Novo Nordisk
REVIEW RETURNED	05-Mar-2019

GENERAL COMMENTS	This is a large cross-sectional study examining the relationship between social media use and sleep. While the study includes a large sample, there are key limitations to the study that make the findings less robust than the authors claim. A key limitation is that the validation of the questions used has not been carried out. It has been established that self-reported measures of sleep are not very robust and can affect the findings. For example, it is very difficult to estimate sleep latency. Categorising wake and sleep times is also problematic. A lot of work has been done in validation of sleep measures, and these need full discussion. The social media use is also problematic; this needs validation. Many subjective factors will bias this. There are already data from UK cohorts that have used more robust measures including prospective studies. These need to be discussed fully and how the current work adds to this. The cross-sectional nature of the study is also problematic and it's difficult for to make causal conclusions. While the study has a large sample, the issues with the methodology make it difficult to use findings as a backbone to inform public health policy.
--

REVIEWER	Stephen H. Sheldon, D.O., F.A.A.P. Northwestern University Feinberg School of Medicine
REVIEW RETURNED	11-Mar-2019

GENERAL COMMENTS	"Social media use and adolescent sleep out comes: cross-sectional findings from the UK Millennium Cohort Study." Manuscript ID: BMJ-2019-049372 Authors: Scott H, Biello S, Woods, H.
--

	This is a novel study utilizing six self-reported sleep parameters quantifying the effect on sleep of high social media users compared to relatively typical users of social media. The study's importance is directly related to policy and health implications derived from a very large cohort of adolescents in the UK, rather than from anecdotal evidence. It also provides for development of normative data and refocusing from correlations between generic terms, such as screen time and well-being, to meaningful outcome measures related to social media use and sleep related complaints. Methodology provides for scientific validity and reliability. If sample size can be matched, it can provide for reproducibility. The study principally provides empirical evidence from which sound public policy decisions can be made. The research question is clearly defined. Methodology provides for appropriate answers due to an exquisite statistical analysis. Study design is well thought out, clearly presented, and it appears adequate to answer proposed questions. Participants were part of the UK Millennium Cohort Study (UKMCS). The cohort was very adequately described, and conditions defined. Although the participants were part of this extensive study, there likely should be some statement regarding how this study was conducted, how consent was obtained, and whether there were inclusion and exclusion criteria. This would be important to readers who are not familiar with the UKMCS. Data presented clearly answers questions proposed. Data and tables were credible and well presented. This reviewer would like to know a bit more about the data-base from which these data were obtained/extracted. Nonetheless, data interpretation was clear and well presented. Conclusions were based on the data. Limitations of the study were accurately and clearly defined, with recommendations for future research. References were up-to-date and relevant. There were no significant omissions. Abstract, summary and key message clearly focuses on what these data add and reflects accurately what the paper presents. Strengths 1. Large sample size.2. Sound methodology and statistical analysis.3. Conclusions are clearly based on data.4. Listing of limitation of the study and recommendations for future research are clear.5. This paper is very well written and presented. It is easy to follow and appropriate for general audiences.6. The study has significant importance for public policy development based on empirical data rather than anecdotes.7. Overall, this is an outstanding and important study. Weaknesses 1. This reviewer would like to see more information about the UKMCS.
--	--

	2. Information regarding the subjective nature of the data should be stressed. 3. There is lack of validation of the questions asked on the UKMCS. However, this was clearly pointed out in the limitations. Nevertheless, this limitation should be pointed out in the analysis within this context. 4. Questions regarding sleep latency is likely the softest finding since misperception of sleep state is not uncommon (Fernandez-Mendoza J, et.al. Sleep Misperception and Chronic Insomnia in the General Population: Role of Objective Sleep Duration and Psychological Profiles. Psychosomatic Medicine 2011; 73(1): 88-97). The concept of total sleep time is also only briefly addressed. This might be somewhat expanded. 5. The most significant question raised is utilization of the term, "Sleep Outcomes." This term might suggest objective information for which there is no validation. Consideration might be given to changing the term to "Reported Sleep Patterns" or "Reported Sleep Habits." Thank you very much for allowing me to review this excellent manuscript.
--	---

REVIEWER	Mukesh Kapoor Rochester Regional Health, Rochester, NY, USA
REVIEW RETURNED	25-Mar-2019

GENERAL COMMENTS	I wanted to thank the authors for their very interesting and well researched article. It certainly adds more information to our understanding around social media use in children. I had a few questions/suggestions:  1. Two of the survey questions ask "What time do you usually go to sleep" respectively on a school night and when you do not have school the next day. In clinic, we commonly tend to see patients respond to "What time do you usually go to sleep" in different ways. Some patients interpret "What time do you usually go to sleep" as the actual time they fall asleep. However, other patients interpret this as the time they go to bed (but not fall asleep). This can also be seen (but to a lesser extent) with patient reported wake time. For some, this may mean the time that sleep actually ends and when their eyes open for good, but for others, this may mean when they actually get out of bed (but may have awoken earlier and were laying in bed awake). The authors could consider discussing these issues further in their discussion. 2. The term "typical user" and "average user" seem to be used interchangeably. It may make for easier reading by simply using the term "average user". 3. In certain areas of the paper, average use is defined as 1 hour to less than 3 hours, high use as 3 hours to less than 5 hours, and very high use to 5 hours or more. However, in other areas of the paper, average use is defined as 1-3 hours, high use as 3-5 hours and very high use as 5+ hours. Can the authors be more precise about their definitions? 4. Female subjects in this study had more high and very high use compared to male subjects. Can the authors hypothesize the possible reason behind this? Should interventions around social media use take these findings in to account?
---

	5. Do the authors have any data around social media use on school days vs. free days? 6. In the paper, the term "frequent night time awakenings" seems to have been used to describe night time awakenings after which individuals had trouble falling back asleep. However, literally speaking, frequent night time awakenings signifies multiple night time awakenings. Some people may have no trouble falling back asleep after these awakenings and other people may have a hard time falling back asleep after these awakenings. Similarly, there can be patients who have only a single night time awakening and have no trouble going back to sleep after this and other patients who can have a hard time falling back asleep after this single nighttime awakening. The questionnaire asks "How often did you awaken during your sleep time and have trouble falling back to sleep again?". Perhaps in the paper, the authors may want to use the term "trouble falling back asleep after a night time awakening" rather than the term "frequent night time awakening". 7. Did the authors find any demographic, social, BMI etc. differences between high and very high users compared to low and average users? 8. Did the survey have any questions re: sleep disorders, snoring etc.? 9. Page 11 - first paragraph - second line - "and between 11 pm and midnight on free days (with 36% falling asleep later; see Table 1)". In table 1, this number appears to be 34%. Please clarify. 10. Page 14 - second paragraph - fourth line: "after 11 pm on school days and free days". Should this read "after 11 pm on school days and after midnight on free days"? Please clarify. 11. On page 15 - second paragraph - first line - "Delayed sleep onset on school nights is therefore a key issue to target". I think that the authors should make the case for delayed sleep onset on both school and free nights. In clinic, we often see that by sleeping in on their off days, patients have trouble going to sleep in the night and then have to wake up early the next day as the school/work week starts. This makes them feel sleepy and tired in the daytime. Thus, the target (though likely hard to achieve) could be a more consistent, non-delayed sleep schedule throughout the week. A better target might be as outlined in # 12 below. 12. Supplementary Tables 1 -4: On school days, the majority of the children seem to be going to sleep before 11 pm and waking up before 8 am. However, on free days, the majority of the children seem to be going to sleep after 11 pm and waking up after 8 am. Such significant changes in the sleep schedule on the free days may reflect the biological delay that occurs in this age group and should be reason for policy makers to consider delayed school times. The authors could consider discussing this further.
--	--

REVIEWER	Patrick Archambault Université Laval
REVIEW RETURNED	02-Apr-2019

GENERAL COMMENTS	I would like to thank the authors for allowing me to review their important work.
---

	The question addressed by the authors in this paper is an important one that is a rising issue in millennial adolescents. I particularly like how the authors investigate the specific role of social media vs generic screen time exposure. With their results, the authors point to interesting future investigations to further advance the understanding of the impact of social media overuse on sleep in adolescents. I have a few comments that I think would need to be addressed: 1- I question the use of self-reported sleep outcomes. These outcomes are exposed to bias. The authors acknowledge this in their limitations, but I would have liked to know if previous studies have studied these self-report outcomes in relation to more objectively measured outcomes. Are they well correlated in other studies? 2- What is the risk for having adjusted for covariates that are potentially a consequence of poor sleep, such as wellbeing, depression and anxiety? 3- Previous studies have demonstrated that later sleep predicts poorer academic and emotional outcomes, but do the authors have any outcomes that could verify this relationship in their large cohort? These analyses could be important to try to establish if social media has a negative impact on school performance and emotional outcomes as well as sleep disturbances. 4- Are there any positive impacts of social media use among adolescents that could be measured in this large cohort that could present a balanced view of the use of social media use in adolescents? For example, sense of belonging, decreased loneliness? Where any of these measures performed in this cohort study?
--	--

VERSION 1 – AUTHOR RESPONSE

Reviewer 1

This is a large cross-sectional study examining the relationship between social media use and sleep. While the study includes a large sample, there are key limitations to the study that make the findings less robust than the authors claim. A key limitation is that the validation of the questions used has not been carried out. It has been established that self-reported measures of sleep are not very robust and can affect the findings. For example, it is very difficult to estimate sleep latency. Categorising wake and sleep times is also problematic. A lot of work has been done in validation of sleep measures, and these need full discussion.

- Thank you for raising the issue of the accuracy of self-report estimates of sleep parameters. We have extended our Limitations section to further discuss this point (pp. 17-18). In this section, we now make it clearer to the reader that “the current self-reported sleep measures offer valuable insight into adolescents’ subjective experience of sleep”. We highlight that this analysis of self-report sleep measures provides one part of the picture and that there is a continued need to triangulate insight from multiple methodologies, including objective sleep measures. We also now explicitly discuss sleep state misperception as a common limitation of self-report estimate accuracy of sleep parameters.

- As well as discussing this in our Limitations section, as outlined above, we now also highlight this point in our Methods section (under Materials, pp. 7-8). This ensures that the reader is aware that these are unvalidated single-item self-report survey questions.

The social media use is also problematic; this needs validation. Many subjective factors will bias this.

- We agree that limiting measurement of social media use to a single unvalidated question of typical hours per day is problematic, and this highly prevalent approach is one key limitation of available literature as a whole. We acknowledge the value of the UK Millennium Cohort Study usefully providing a specific measure of social media use (separate from other technology use). However, we strongly feel that a key priority for this field is a focus on rigorously developing and validating high quality measurement tools that more meaningfully capture a range of social media experiences. We have now articulated this point in our Limitations section (p. 17), when highlighting the limitations of a single unvalidated item to measure social media.

There are already data from UK cohorts that have used more robust measures including prospective studies. These need to be discussed fully and how the current work adds to this.

- Thank you for raising this question. We note that sleep has historically been overlooked in public health and education, with increasing recent recognition of its crucial role in health and wellbeing (p. 4). Therefore, not all cohort studies routinely assess sleep as a measure of health and wellbeing. We also note that social media is often measured only grouped together with other technologies under generic measures of “screen time” or “technology use” (p. 5). As such, after reviewing available data sources, we identified the UK Millennium Cohort Study as the appropriate representative UK adolescent dataset to answer the current research question. It includes both a measure of social media use specifically (not generic phone use, or technology use), and a range of sleep parameters to capture the adolescent experience of sleep timing (on both school days and free days) and sleep quality. We have added further details on the UK Millennium Cohort Study (p. 7), noting that it provides the appropriate measures (a range of sleep habits and social media use) in order to answer the current research question (pp. 7-8).

The cross-sectional nature of the study is also problematic and it's difficult for to make causal conclusions.

- Thank you for this comment. We have now extended our discussion of the limitations of the current cross-sectional analyses (p. 18). We explicitly state that this precludes causal conclusions. We note recent examples of longitudinal and experimental work that is adding to current available (mostly cross-sectional) evidence. Despite the use of cross-sectional data, the current analyses do address a number of existing gaps in current evidence, as outlined in the introduction and conclusions. This study can therefore usefully inform future efforts to conduct longitudinal and experimental work, by first filling these gaps in understanding.

While the study has a large sample, the issues with the methodology make it difficult to use findings as a backbone to inform public health policy.

- Thank you for taking the time to provide constructive feedback on our manuscript. Making the revisions as outlined above has improved the clarity of our manuscript in expressing its contributions and limitations. This study clearly identified gaps in current evidence: a lack of social media-specific evidence; a lack of comprehensive covariate in previous studies; a need for normative baseline data profile to make meaningful comparisons. It then makes use of an appropriate available data source to address these gaps in a large representative UK sample. We explicitly discuss the limitations of this one study, positioning it as one piece of evidence that is available to help inform ongoing discussions and debate in policy and practice. We outline directions for future work to further enhance available evidence and tools.

Reviewer 2

This is a novel study utilizing six self-reported sleep parameters quantifying the effect on sleep of high social media users compared to relatively typical users of social media. The study's importance is directly related to policy and health implications derived from a very large cohort of adolescents in the UK, rather than from anecdotal evidence. It also provides for development of normative data and refocusing from correlations between generic terms, such as screen time and well-being, to meaningful outcome measures related to social media use and sleep related complaints.

Methodology provides for scientific validity and reliability. If sample size can be matched, it can provide for reproducibility. The study principally provides empirical evidence from which sound public policy decisions can be made.

The research question is clearly defined. Methodology provides for appropriate answers due to an exquisite statistical analysis. Study design is well thought out, clearly presented, and it appears adequate to answer proposed questions.

Participants were part of the UK Millennium Cohort Study (UKMCS). The cohort was very adequately described, and conditions defined. Although the participants were part of this extensive study, there likely should be some statement regarding how this study was conducted, how consent was obtained, and whether there were inclusion and exclusion criteria. This would be important to readers who are not familiar with the UKMCS.

Data presented clearly answers questions proposed. Data and tables were credible and well presented. This reviewer would like to know a bit more about the data-base from which these data were obtained/extracted. Nonetheless, data interpretation was clear and well presented. Conclusions were based on the data. Limitations of the study were accurately and clearly defined, with recommendations for future research.

References were up-to-date and relevant. There were no significant omissions.

Abstract, summary and key message clearly focuses on what these data add and reflects accurately what the paper presents.

Strengths

1. Large sample size.
2. Sound methodology and statistical analysis.
3. Conclusions are clearly based on data.
4. Listing of limitation of the study and recommendations for future research are clear.
5. This paper is very well written and presented. It is easy to follow and appropriate for general audiences.
6. The study has significant importance for public policy development based on empirical data rather than anecdotes.
7. Overall, this is an outstanding and important study.

- Thank you for taking the time to provide this comprehensive and constructive feedback on the strengths and weaknesses of our manuscript. We have addressed each of the weaknesses you raised below, which has allowed us to improve the clarity of our manuscript in presenting the contribution and limitations of this study.

Weaknesses

1. This reviewer would like to see more information about the UKMCS.
 - We have now added more information to the Methods section (under 'Participants', p. 7). This outlines the purpose of the cohort study, previous survey sweeps, how the interviews were conducted and consent processes for parents and young people.
2. Information regarding the subjective nature of the data should be stressed.
 - Thank you for highlighting this point. We have now stressed the strengths and limitations of subjective self-report on sleep and social media use more clearly in the 'Limitations' section of the Discussion (pp. 17-18). It notes that the available measures offer insight into adolescents' subjective experience of sleep and social media use, but that these estimates can diverge from more objective measures. This is further discussed in the context of sleep state misperception regarding sleep onset latency, in response to your comment number 4.

3. There is lack of validation of the questions asked on the UKMCS. However, this was clearly pointed out in the limitations. Nevertheless, this limitation should be pointed out in the analysis within this context.

- As well as discussing this in our Limitations section (p. 17), we now also raise this point in our Methods section (under Materials, pp. 7-8). This ensures that the reader is aware that these are unvalidated single-item survey questions.

4. Questions regarding sleep latency is likely the softest finding since misperception of sleep state is not uncommon (Fernandez-Mendoza J, et.al. Sleep Misperception and Chronic Insomnia in the General Population: Role of Objective Sleep Duration and Psychological Profiles. *Psychosomatic Medicine* 2011; 73(1): 88-97). The concept of total sleep time is also only briefly addressed. This might be somewhat expanded.

- Thank you for this constructive recommendation. We now explicitly discuss sleep state misperception as a common limitation of self-report estimate accuracy of sleep parameters (pp. 17-18). This section now makes it clearer to the reader that “the current self-reported sleep measures offer valuable insight into adolescents’ subjective experience of sleep”. We highlight that this analysis of self-report sleep measures provides one part of the picture and that there is a continued need to triangulate insight from multiple methodologies, including objective sleep measures.

- Regarding total sleep time, the available response categories for sleep onset (e.g. 9–10 pm) and wake times (e.g. 7–8 am) meant that it was unfortunately not possible to calculate exact total sleep times, or to accurately order participants by sleep time due to overlapping bands of possible sleep times (e.g. 8-10h, 9-11h). However, we do address the typical pattern of shortened sleep opportunity with relatively stable weekday rise times, meaning that late sleep onset is a good proxy for short sleep (p. 15).

5. The most significant question raised is utilization of the term, “Sleep Outcomes.” This term might suggest objective information for which there is no validation. Consideration might be given to changing the term to “Reported Sleep Patterns” or “Reported Sleep Habits.”

- Thank you for this recommendation. We now use “sleep patterns” as opposed to “sleep outcomes” throughout the manuscript, for increased clarity. As noted above, we also attend more fully to the limitations of self-reported estimates of sleep patterns.

Thank you very much for allowing me to review this excellent manuscript.

Reviewer 3

I wanted to thank the authors for their very interesting and well researched article. It certainly adds more information to our understanding around social media use in children. I had a few questions/suggestions:

- Thank you for your detailed and constructive comments on our manuscript. We have addressed these point by point below, noting the changes we have now made to improve our manuscript.

1. Two of the survey questions ask "What time do you usually go to sleep" respectively on a school night and when you do not have school the next day. In clinic, we commonly tend to see patients respond to "What time do you usually go to sleep" in different ways. Some patients interpret "What time do you usually go to sleep" as the actual time they fall asleep. However, other patients interpret this as the time they go to bed (but not fall asleep). This can also be seen (but to a lesser extent) with patient reported wake time. For some, this may mean the time that sleep actually ends and when their eyes open for good, but for others, this may mean when they actually get out of bed (but may have awoken earlier and were laying in bed awake). The authors could consider discussing these issues further in their discussion.

- Thank you for raising this relevant point. We have now noted that some participants may have reported their bedtime (as opposed to sleep onset time), in which case their sleep onset time would be even further delayed (p. 14). This includes a citation to recent work (Exelmans & Van den Bulck, 2017) that discusses this issue further.

2. The term "typical user" and "average user" seem to be used interchangeably. It may make for easier reading by simply using the term "average user".

- Thank you for this suggestion. We have now used "average user" throughout the manuscript to improve readability.

3. In certain areas of the paper, average use is defined as 1 hour to less than 3 hours, high use as 3 hours to less than 5 hours, and very high use to 5 hours or more. However, in other areas of the paper, average use is defined as 1-3 hours, high use as 3-5 hours and very high use as 5+ hours. Can the authors be more precise about their definitions?

- Thank you for pointing out this inconsistency. We have now ensured consistency throughout the tables and text, using more precise labels (i.e. "1 to <3 hours", "3 to <5 hours").

4. Female subjects in this study had more high and very high use compared to male subjects. Can the authors hypothesize the possible reason behind this? Should interventions around social media use take these findings in to account?

- Thank you for highlighting this point for further discussion. We have now extended the first paragraph of our Discussion (pp. 13-14) to highlight this gender difference. Similar gender patterns have been noted in previous studies (now cited in text), but there has been little work that can yet support an evidence-based explanation for this difference. Therefore, we have noted this as an interesting area for continued research: "to explore the sleep implications of how adolescent boys and girls spend their time on social media (with previous evidence of gender differences in preferred platforms, motivations and self-presentation)."

5. Do the authors have any data around social media use on school days vs. free days?

- We agree that data on school vs. free day social media use would be interesting. However, the cohort members were only asked a single question on social media use, which referred to a typical school day. We have extended our Limitations section (p. 17) to further discuss the importance of developing and validating measures that provide a more holistic understanding of a range of social media habits and experiences.

6. In the paper, the term "frequent night time awakenings" seems to have been used to describe night time awakenings after which individuals had trouble falling back asleep. However, literally speaking, frequent night time awakenings signifies multiple night time awakenings. Some people may have no trouble falling back asleep after these awakenings and other people may have a hard time falling back asleep after these awakenings. Similarly, there can be patients who have only a single night time awakening and have no trouble going back to sleep after this and other patients who can have a hard time falling back asleep after this single nighttime awakening. The questionnaire asks "How often did you awaken during your sleep time and have trouble falling back to sleep again?". Perhaps in the paper, the authors may want to use the term "trouble falling back asleep after a night time awakening" rather than the term "frequent night time awakening".

- Thank you for this recommendation. We agree that this rewording provides more clarity. We have now changed "frequent nighttime awakenings" to "trouble falling back asleep after nighttime awakening" throughout the manuscript.

7. Did the authors find any demographic, social, BMI etc. differences between high and very high users compared to low and average users?

- Thank you for suggesting this point. As well as the gender breakdown of social media use (reported in Table 1, p. 9), we have now added a breakdown according to other demographics (ethnicity and household income) to the Supplementary Materials. This additional material is signposted in the Notes below Table 1 for interested readers. This shows that non-white cohort members were more likely to be low social media users, and that the prevalence of very high use decreased with household income.

8. Did the survey have any questions re: sleep disorders, snoring etc.?

- We agree that questions on other sleep complaints would offer interesting insight. However, the six sleep parameters analysed here were the only questions that the cohort answered on sleep.

9. Page 11 - first paragraph - second line - "and between 11 pm and midnight on free days (with 36% falling asleep later; see Table 1)". In table 1, this number appears to be 34%. Please clarify.

- Thank you. We have corrected the in-text typo to the correct value: 34% (p. 10).
10. Page 14 - second paragraph - fourth line: "after 11 pm on school days and free days". Should this read "after 11 pm on school days and after midnight on free days"? Please clarify.
- Thank you for pointing this out. We have now corrected this to read "later than average, i.e. after 11pm on school days and after midnight on free days." (p. 14)
11. On page 15 - second paragraph - first line - "Delayed sleep onset on school nights is therefore a key issue to target". I think that the authors should make the case for delayed sleep onset on both school and free nights. In clinic, we often see that by sleeping in on their off days, patients have trouble going to sleep in the night and then have to wake up early the next day as the school/work week starts. This makes them feel sleepy and tired in the daytime. Thus, the target (though likely hard to achieve) could be a more consistent, non-delayed sleep schedule throughout the week. A better target might be as outlined in # 12 below.
- Thank you for this suggestion. We have adjusted our wording here to propose working towards "an appropriate and consistent sleep schedule across the week, particularly to allow sufficient sleep on school nights". (p. 15) This highlights the need for consistent routine across the week, which should in turn offer particular benefits to school night sleep duration, since fixed rise times mean that delayed bedtimes equate to shorter sleep opportunity.
12. Supplementary Tables 1 -4: On school days, the majority of the children seem to be going to sleep before 11 pm and waking up before 8 am. However, on free days, the majority of the children seem to be going to sleep after 11 pm and waking up after 8 am. Such significant changes in the sleep schedule on the free days may reflect the biological delay that occurs in this age group and should be reason for policy makers to consider delayed school times. The authors could consider discussing this further.
- Thank you for raising this relevant point. We have extended our point regarding sleep restriction from school rise times in our Discussion (p. X). We now explicitly state that: "Across the sample, this observed pattern of later sleep onset and rise times on free days compared to school days is consistent with the delaying circadian rhythm during this developmental period^{1, 2}, with increasing pressure on policymakers to delay school start times to better align with adolescent body clocks³." (p. 15)

Reviewer 4

I would like to thank the authors for allowing me to review their important work.

The question addressed by the authors in this paper is an important one that is a rising issue in millennial adolescents. I particularly like how the authors investigate the specific role of social media vs generic screen time exposure. With their results, the authors point to interesting future investigations to further advance the understanding of the impact of social media overuse on sleep in adolescents.

- Thank you for taking the time to review our work and provide these constructive comments to improve our manuscript. We have addressed these below, noting the updates we have made to our manuscript to clarify the scope, contribution and limitations of our approach and to highlight interesting avenues for future work to build on the current findings.

I have a few comments that I think would need to be addressed:

1- I question the use of self-reported sleep outcomes. These outcomes are exposed to bias. The authors acknowledge this in their limitations, but I would have liked to know if previous studies have studied these self-report outcomes in relation to more objectively measured outcomes. Are they well correlated in other studies?

- Thank you for raising this question. We have expanded our discussion of this point in our Limitations section (pp. 17-18). We note that these analyses make use of available large-scale representative data, whose self-report nature offers useful insight into the subjective experience of sleep and social media. We have highlighted that this comes with the limitation of lower accuracy of

sleep state perception, which can result in subjective and objective measures of sleep patterns diverging (particularly for sleep onset latency). Therefore, we highlight that this analysis of self-report sleep measures provides one part of the picture but that there is a continued need to triangulate insight from multiple methodologies, including objective sleep measures.

2- What is the risk for having adjusted for covariates that are potentially a consequence of poor sleep, such a wellbeing, depression and anxiety?

- Thank you for this interesting question. We have now clarified within the Discussion that we used these wellbeing measures as covariates given the purpose of the current study to isolate associations between social media use and different sleep measures, and to identify which persisted independent of covariates like wellbeing (p. 16). We now highlight “the increasing recognition of sleep and mental health as two inextricably linked aspects of health”, and propose more complex model testing (especially on longitudinal data) as a future avenue to better understand the complex “likely bidirectional and interactive effects between social media use, sleep, mental health and other associated measures”.

3- Previous studies have demonstrated that later sleep predicts poorer academic and emotional outcomes, but do the authors have any outcomes that could verify this relationship in their large cohort? These analyses could be important to try to establish if social media have a negative impact on school performance and emotional outcomes as well as sleep disturbances.

- Thank you for highlighting this interesting area for future work. The contribution of the current study is to isolate and quantify associations between social media use and sleep, since it is a key component of health and functioning. We agree that examining how sleep may mediate links between social media use and school- or wellbeing- outcomes is an interesting avenue for future work. However, we feel that this is beyond the scope of the current study, which first seeks to address key gaps in existing understanding of social media and sleep (which is in itself a highly discussed topic, with calls for evidence-based policy and practice recommendations). We do now note the valuable potential for future work to examine the complex mechanisms linking “social media use, sleep, mental health and other associated measures, such as school performance” (p. 16).

4- Are there any positive impacts of social media use among adolescents that could be measured in this large cohort that could present a balanced view of the use of social media use in adolescents? For example, sense of belonging, decreased loneliness? Where any of these measures performed in this cohort study?

- Thank you for raising this interesting point. The purpose of this study is to isolate and quantify links between social media use and a range of sleep measures. Therefore, whilst we agree that examining other measures (e.g. sense of belonging or decreased loneliness) could provide interesting insight, we feel that this is beyond the scope of the current study. We agree that a balanced view of social media and adolescent wellbeing is important and have now communicated this more clearly, including the following updates:

- We have highlighted the potential for future studies to use complex model testing to “build a more holistic and balanced understanding of social media’s links to both positive and negative aspects of health and wellbeing” simultaneously (p. 16).

- We have discussed the need to develop and validate more holistic measures of social media experiences as a key priority for the field moving forward (p. 17). This will support future efforts to provide “a more balanced view of both positive and negative impacts of social media experiences”. This is especially true since positive impacts are often not associated with time spent using social media, but instead with more nuanced measures of how individuals engage and interact via these platforms (e.g. whether they experience a meaningful social connection).

VERSION 2 – REVIEW

REVIEWER	Shahrad Taheri Weill Cornell Medicine Qatar
REVIEW RETURNED	19-Jun-2019

GENERAL COMMENTS	The authors have addressed all comments from the BMJ review. In particular, they have acknowledged the key limitations of their study and adjusted the manuscript accordingly.
--

REVIEWER	Patrick Archambault Université Laval, Canada
-----------------	---

REVIEW RETURNED	01-Jul-2019
-------------

GENERAL COMMENTS	I thank the authors for addressing my questions adequately.
---